# Machine-learning-assisted insight into spin ice $Dy_2Ti_2O_7$

Anjana M. Samarakoon [1✉], Kipton Barros[2], Ying Wai Li[3], Markus Eisenbach[3,4], Qiang Zhang[1,5], Feng Ye[1], V. Sharma[6], Z.L. Dun [6], Haidong Zhou[6], Santiago A. Grigera[7,8], Cristian D. Batista [1,6] & D. Alan Tennant [4]

Complex behavior poses challenges in extracting models from experiment. An example is spin liquid formation in frustrated magnets like $Dy_2Ti_2O_7$. Understanding has been hindered by issues including disorder, glass formation, and interpretation of scattering data. Here, we use an automated capability to extract model Hamiltonians from data, and to identify different magnetic regimes. This involves training an autoencoder to learn a compressed representation of three-dimensional diffuse scattering, over a wide range of spin Hamiltonians. The autoencoder finds optimal matches according to scattering and heat capacity data and provides confidence intervals. Validation tests indicate that our optimal Hamiltonian accurately predicts temperature and field dependence of both magnetic structure and magnetization, as well as glass formation and irreversibility in $Dy_2Ti_2O_7$. The autoencoder can also categorize different magnetic behaviors and eliminate background noise and artifacts in raw data. Our methodology is readily applicable to other materials and types of scattering problems.

[1] Neutron Scattering Division, Oak Ridge National Laboratory, 1 Bethel Valley Road, Oak Ridge, TN 37831, USA. [2] Theoretical Division and CNLS, Los Alamos National Laboratory, Los Alamos, NM 87545, USA. [3] National Center for Computational Sciences, Oak Ridge National Laboratory, 1 Bethel Valley Road, Oak Ridge, TN 37831, USA. [4] Materials Science and Technology Division, Oak Ridge National Laboratory, 1 Bethel Valley Road, Oak Ridge, TN 37831, USA. [5] Department of Physics and Astronomy, Louisiana State University, Baton Rouge, LA 70803, USA. [6] Department of Physics and Astronomy, University of Tennessee, Knoxville, TN 37996, USA. [7] Instituto de Física de Líquidos y Sistemas Biológicos, UNLP-CONICET, La Plata, Argentina. [8] School of Physics and Astronomy, University of St Andrews, St Andrews, UK. ✉email: samarakoonam@ornl.gov

Extracting the correct interactions from experimental data is essential for modeling. For magnetic insulators, the model is described by the spin Hamiltonian equation, dictated by symmetry, single-ion properties, and electron overlap between ions. The problem of extracting a spin Hamiltonian from neutron scattering data (inverse scattering problem) is often ill-posed and compounded by the need to use theory to interpret scattering data. Further, the available experimental data may not be enough to accurately determine the model parameters because of limited access to experimental data, a large noise magnitude at each scattering wavevector, or systematic errors associated with, e.g., background subtraction. Selecting the optimal Hamiltonian to model the experimental data is often a formidable task, especially when many parameters must be simultaneously determined. Tools for doing so are needed to uncover the physics that is emerging from large classes of complex magnetic materials[1,2].

$Dy_2Ti_2O_7$ is a highly frustrated magnet showing complex behavior including spin liquid formation[3–8]. The magnetism originates from $Dy^{3+}$ ions which behave as classical Ising spins on a pyrochlore lattice of corner-linked tetrahedra, as in Fig. 1a. Figure 1b shows the four essential magnetic interactions including a ferromagnetic coupling that results from the combination of exchange with a large dipolar interaction. This FM coupling makes $Dy_2Ti_2O_7$ a canonical spin-ice material, i.e., the spins on each tetrahedron obey the ice rules that only allow for two-in two-out configurations[9,10]. This divergenc-free condition leads to a spin liquid with macroscopic degeneracy that features north and south charged magnetic monopoles interacting via a $1/r$ potential at elevated temperatures[3]. A full low temperature characterization demands the study of a vast number of spins subject to short and long-range interactions. Spin dynamics occurs through millisecond quantum tunneling processes[11] and the measured characteristic equilibration time $\tau$ increases drastically upon lowering the temperature, leading to irreversible behavior below 600 mK[12–15]. This slowdown has resulted in major difficulties[16,17] in measuring and interpreting experiments such as heat capacity at low temperatures.

Here we introduce an autoencoder-based approach that can potentially address important modeling challenges, such as a proper background and noise subtraction, more reliable inference of model Hamiltonians, improved transferability to other physical systems, and efficiency. We apply our method to neutron scattering measurements of $Dy_2Ti_2O_7$ in order to infer the optimal parameters for a dipolar spin ice model description.

## Results

**Neutron scattering measurements and simulations.** Here we use diffuse neutron scattering from time-of-flight techniques (see

Methods: Experimental details) on the CORELLI instrument at the Spallation Neutron Source, Oak Ridge National Laboratory to measure the magnetic state of $Dy_2Ti_2O_7$. Three-dimensional (3D) volumes of diffuse scattering were measured in the 100–960 mK temperature range. In view of the low temperature equilibration challenge, we undertake our analysis on data sets at 680 mK, which is low enough for correlations to be well developed but sufficiently high to reach equilibrium over a short time scale. Figure 2a shows the background-subtracted data at 680 mK. This is proportional to the modulus squared of the spin components in the wavevector space. However, an additional aspect of neutron scattering is that it samples only the spin components perpendicular to the wavevector transfer $\mathbf{Q}$.

We employ a dipolar spin-ice Hamiltonian that includes exchange terms up to third-nearest neighbors:

$$H = J_1 \sum_{\langle i,j \rangle_1} \mathbf{S}_i \cdot \mathbf{S}_j + J_2 \sum_{\langle i,j \rangle_2} \mathbf{S}_i \cdot \mathbf{S}_j + J_3 \sum_{\langle i,j \rangle_3} \mathbf{S}_i \cdot \mathbf{S}_j$$
$$+ J_{3'} \sum_{\langle i,j \rangle_{3'}} \mathbf{S}_i \cdot \mathbf{S}_j + Dr_1^3 \sum_{i,j} \left[ \frac{\mathbf{S}_i \cdot \mathbf{S}_j}{|\mathbf{r}_{ij}|^3} - \frac{3(\mathbf{S}_i \cdot \mathbf{r}_{ij})(\mathbf{S}_j \cdot \mathbf{r}_{ij})}{|\mathbf{r}_{ij}|^5} \right] \quad (1)$$

where $\mathbf{S}_i$ can be viewed as an Ising spin of the $i$th ion, Fig. 1b. The model includes first, second, and two different third nearest neighbor interaction strengths, $J_1$, $J_2$, $J_3$, and $J_{3'}$, respectively. There is also a dipolar interaction with strength $D$, which couples the $i$th and the $j$th spins, according to their displacement vector $\mathbf{r}_{ij}$. Prior work has determined $D = 1.3224$ K and $J_1 = 3.41$ K to high accuracy[17–21]. In the present modeling effort, we seek to determine the three unknown parameters $J_2$, $J_3$, and $J_{3'}$ without any use of prior knowledge. Given a model Hamiltonian $H$, we use Metropolis Monte Carlo to generate a simulated structure factor, $S^{sim}(\mathbf{Q})$, to be compared with the experimental data $S^{exp}(\mathbf{Q})$ (Methods: Simulation details).

**Optimizing Hamiltonian for diffuse scattering data.** In a direct approach, one might try to minimize the squared distance,

$$\chi^2_{S(\mathbf{Q})} = \frac{1}{N} \sum_{\mathbf{Q}} m(\mathbf{Q}) \left( S^{exp}(\mathbf{Q}) - S^{sim}(\mathbf{Q}) \right)^2 \quad (2)$$

between the raw experiment and simulation data. We introduce a factor $m(\mathbf{Q}) \in \{0, 1\}$ masking selected $\mathbf{Q}$-points where experimental artifacts can be identified (see Supplementary Fig. 4). The number of non-masked $\mathbf{Q}$-points is $N = \sum_{\mathbf{Q}} m(\mathbf{Q}) \approx 1.2 \times 10^5$. We initially investigated optimization methods, such as the particle swarm method[22], to overcome local barriers and find the global minimum of $\chi^2_{S(\mathbf{Q})}$. However, despite nominal success in optimization, we quickly ran into reliability issues stemming from

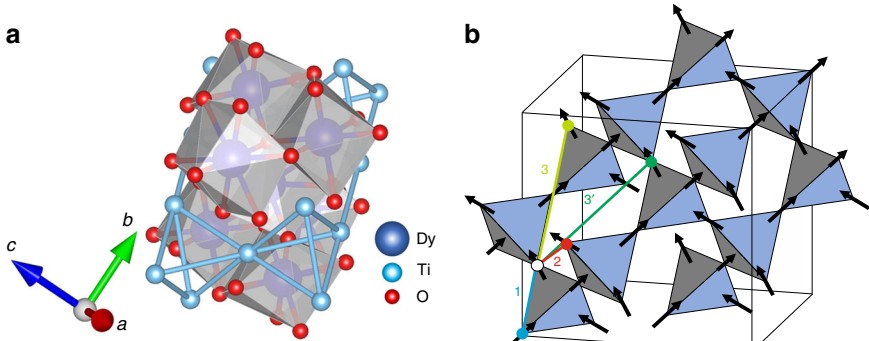

**Fig. 1 Crystal structure and the effective magnetic model. a** Atomic structure of $Dy_2Ti_2O_7$ comprised of tetrahedra of magnetic Dy ions (blue) and nonmagnetic octahedra of oxygen ions (red) surrounding Ti ions (cyan). **b** The magnetic moments located on Dy ions are constrained by crystal field interactions to point in or out of the tetrahedra. They form a corner sharing pyrochlore lattice. The pathways of nearest neighbor (1), next-nearest-neighbor (2) and two inequivalent next-next-nearest neighbor (3 and 3′) interactions are shown as thick colored lines.

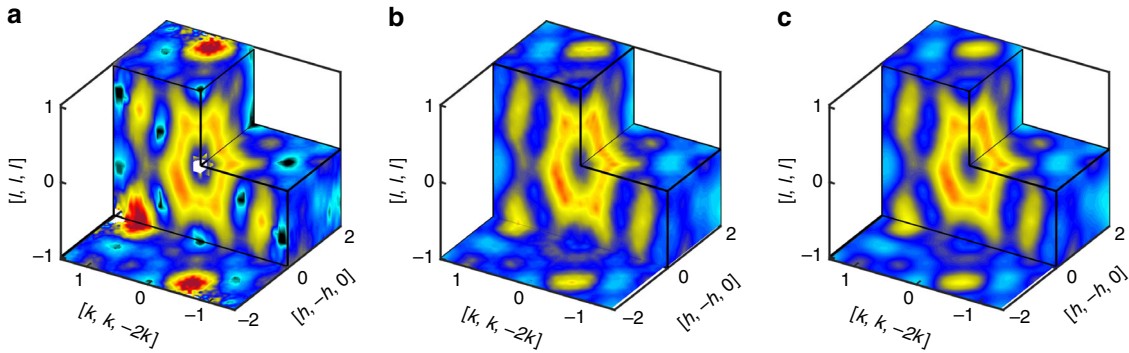

**Fig. 2 Comparison of the experimental and simulated data. a** The scattering function $S^{exp}(\mathbf{Q})$ for the $Dy_2Ti_2O_7$ crystal at 680 mK. **b** The same experimental data after being filtered by the autoencoder, $S^{exp}_{AE}(\mathbf{Q})$. **c** Simulated scattering data from the optimal model spin Hamiltonian, $S^{sim}_{opt}(\mathbf{Q})$. In going from (**a**) to (**b**), the autoencoder has filtered out experimental artifacts such as the red peaks, the missing data at the dark patches, etc. Using both scattering and heat capacity data, we determine the optimal spin Hamiltonian couplings to be $J_2 = 0.00(6)$ K, $J_3 = 0.014$ (16) K and $J_{3'} = 0.096(12)$ K with $J_1 = 3.41$ K and $D = 1.3224$ K having been fixed in prior work[17].

errors in the experimental and simulation data. As we will discuss below, $\chi^2_{S(\mathbf{Q})}$ is both noisy and effectively flat around its minimum, such that many distinct model Hamiltonians could achieve similarly small values of the $\chi^2_{S(\mathbf{Q})}$ error measure. Thus, even if we could find the global minimum of $\chi^2_{S(\mathbf{Q})}$, it might still be far from the physically correct model for $Dy_2Ti_2O_7$.

To address the ill-posed nature of this inverse scattering problem, we present two strategies: (1) We employ machine learning techniques to replace $\chi^2_{S(\mathbf{Q})}$ with our error measure $\chi^2_{S_L}$ that is more robust to errors in the experimental and simulation data, and puts more weight on "characteristic features" of the structure factor. (2) Rather than reporting just the single "best" model, we sample from the entire set of Hamiltonian models for which the error measure is below some tolerance threshold. In this way, our method will report not just a model, but also a model uncertainty.

We use an autoencoder[23] to formulate $\chi^2_{S_L}$, our choice of error measure. Autoencoders were originally developed in the context of computer vision, where they are known to be effective at image compression and denoising tasks. Here we apply them to interpret structure factor data, and to disambiguate among many possible solutions of the inverse scattering problem. Our autoencoder is a neural network that takes an $S(\mathbf{Q})$ as input (either simulated or experimental), encodes it into a compressed latent space representation $S_L$, and then decodes to an output $S_{AE}(\mathbf{Q})$ that captures the essence of the input $S(\mathbf{Q})$, while removing irrelevant noise and artifacts.

The autoencoder's latent space $S_L = (S_1, S_2, \ldots S_D)$ provides a low-dimensional characterization of the $S(\mathbf{Q})$ data. The dimension $D$ of the latent space should strike a balance between overfitting and underfitting. Keeping $D$ relatively small limits the autoencoder's ability to fit irrelevant noise in the training data. On the other hand, $D$ should be large enough to allow the autoencoder flexibility to capture physically relevant characteristics in $S(\mathbf{Q})$. We selected $D = 30$ based on the $D$-dependance of $\Delta S(\mathbf{Q})$ (error over the validation dataset). (see Supplementary Fig. 7)

Note that the physical $S(\mathbf{Q})$ data will contain many more scalar components than the 30 available in the latent space. Thus, by design, the autoencoder's output $S_{AE}(\mathbf{Q})$ can only be an approximation to its input $S(\mathbf{Q})$. After proper training, one hopes that the autoencoder will be able to extract the relevant characteristics of a given $S(\mathbf{Q})$, while discarding irrelevant information such as noise and experimental artifacts. The autoencoder determines what information is relevant according to its ability to encode and faithfully decode the training data (Methods: Training details).

We employ the simplest possible autoencoder architecture: a fully-connected neural network with a single hidden layer. The hidden space activations (i.e., the latent space representation) are defined as $S_L = f(\sum_{\mathbf{Q}} W_{L,\mathbf{Q}} m(\mathbf{Q}) S(\mathbf{Q}) + b_L)$, where $S(\mathbf{Q})$ is the input to the autoencoder, and the matrix $W_{L,\mathbf{Q}}$ and vector $b_L$ are to be determined from the machine learning training process. Given simulated structure factor data as input, we interpolate to the experimental $\mathbf{Q}$-points as necessary. The output of the autoencoder is defined as $S_{AE}(\mathbf{Q}) = f(\sum_{L=1}^{30} W'_{\mathbf{Q},L} S_L + b'_{\mathbf{Q}})$, where the new matrix $W'_{\mathbf{Q},L}$ and vector $b'_{\mathbf{Q}}$ are also trainable. We employ the logistic activation function $f(x) = 1/(1 + e^{-x})$ at both layers. This choice guarantees that the output $S_{AE}(\mathbf{Q})$ is non-negative.

Figure 2 illustrates how the trained autoencoder processes the $Dy_2Ti_2O_7$ scattering data. Figure 2a shows the raw experimental data $S^{exp}(\mathbf{Q})$ while Fig. 2b shows how the autoencoder filters the experimental data to produce $S^{exp}_{AE}(\mathbf{Q})$. The autoencoder preserves important qualitative features of the data, while being very effective at removing experimental artifacts. Figure 2c shows the simulated data $S^{sim}_{opt}(\mathbf{Q})$ for the optimal Hamiltonian model $H_{opt}$, without any autoencoder filtering. We will describe later our procedure to determine $H_{opt}$. Note that the best model, $S^{sim}_{opt}(\mathbf{Q})$, is in remarkably good agreement with $S^{exp}_{AE}(\mathbf{Q})$. This agreement is consistent with the fact that the autoencoder was trained specifically to reproduce simulated data.

Figure 3 provides another way to understand how the autoencoder is processing the $S(\mathbf{Q})$ data. In Fig. 3a we show a cross section of $S^{exp}(\mathbf{Q})$ in the high symmetry plane ($[h, -h, 0]$− $[k, k, -2k]$). Figure 3b shows the corresponding simulated data $S^{sim}_{opt}(\mathbf{Q})$ for the optimal model Hamiltonian. Now we perturb $H_{opt}$ to a new model $H_{perturb}$, which keeps all parameters from $H_{opt}$ except modifies $J_2$ from 0.16 K to −0.18 K. Despite the relatively significant change $\Delta J_2 \cong 0.1 J_1$, there is very little change to the structure factor. Indeed, Fig. 3c shows that $\Delta S^{sim} = S^{sim}_{opt}(\mathbf{Q}) - S^{sim}_{perturb}(\mathbf{Q})$ is an order of magnitude smaller than the peaks in $S^{sim}_{opt}(\mathbf{Q})$, and relatively noisy. This illustrates the inherent difficulty of our inverse scattering problem: many $J_2$ values seem to produce similarly good Hamiltonians.

The autoencoder latent space can be effective in extracting and amplifying important $S(\mathbf{Q})$ features that might otherwise be hidden. To show this, we consider the latent space representations $S_L$ and $S'_L$ for $S^{sim}_{opt}(\mathbf{Q})$ and $S^{sim}_{perturb}(\mathbf{Q})$, respectively. We ask: How much does $S_L$ need to be modified toward $S'_L$ in order to capture the important characteristics of $S^{sim}_{perturb}(\mathbf{Q})$, i.e., the perturbations

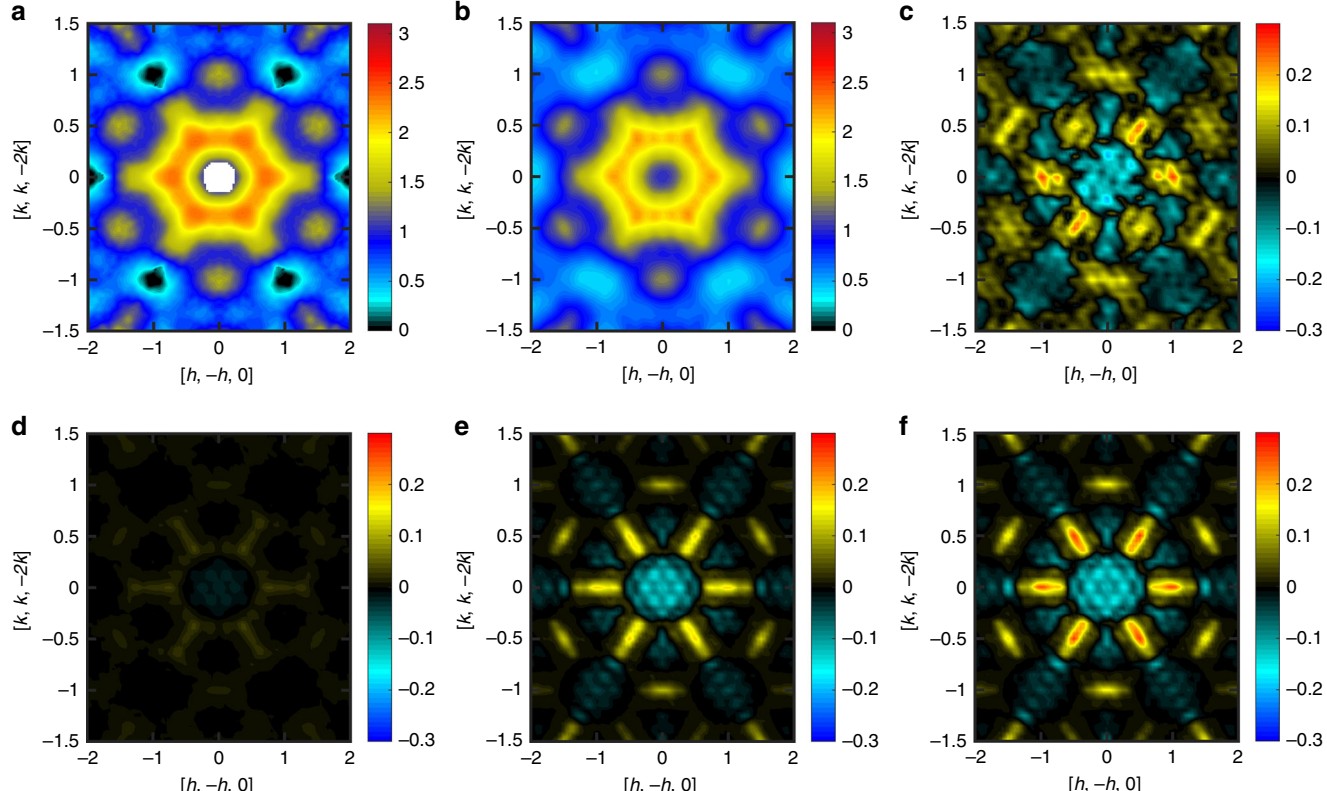

**Fig. 3 Effect of latent space variables on structure factor. a** Experimental scattering data $S^{exp}(\mathbf{Q})$ in the high symmetry plane. Dark patches indicate zero signal due to nuclear Bragg peaks. **b** The corresponding simulated data for the optimal model Hamiltonian. **c** Change in the simulated structure factor $\Delta S^{sim}$ when the $J_2$ coupling is perturbed from 0.16 K to −0.18 K. The relatively weak response $\Delta S^{sim}$ to a perturbation in $J_2$ of ±0.05 $J_1$ illustrates the challenge in inferring the correct spin Hamiltonian. **d**–**f** The change in the autoencoder output when 1, 6, and 12 latent space components, respectively, are updated to account for the perturbation on $J_2$ (see main text for details).

to the structure factor? To answer this question, we replace 1, 6, and 12 latent space components of $S_L$ with the corresponding ones in $S'_L$. The components selected are those with the largest deviations, $|S_L - S'_L|$. Figure 3d–f show the change in autoencoder output, after substitution of the latent space components. Panel f captures some physically important characteristics of $\Delta S^{sim}$ while discarding irrelevant noise. Latent space components beyond 12 carry little information about $\Delta S^{sim}$.

Now we show how the autoencoder can assist in solving the inverse scattering problem, i.e., in finding the optimal model Hamiltonian $H_{opt}$ given the experimental data $S^{exp}(\mathbf{Q})$. To illustrate the important ideas, we first focus on determining $J_2$, assuming the other parameters of $H_{opt}$ are already known.

Figure 4a shows $\chi^2_{S(\mathbf{Q})}$ as a function of $J_2$, illustrating the difficulty in making direct comparisons between experimental and simulated scattering data, Eq. (2). In principle the minimum of $\chi^2_{S(\mathbf{Q})}$ would give $J_2$, but in practice one must contend with relatively large uncertainties in the data. The visible scatter in Fig. 4a is mostly a consequence of limited statistics of the simulated data. Other sources of error, such as systematic experimental error, will also exist and are more difficult to quantify.

A natural modification is to replace $\chi^2_{S(\mathbf{Q})}$ with the squared distance of autoencoder-filtered structure factors,

$$\chi^2_{S_{AE}(\mathbf{Q})} = \frac{1}{N}\sum_{\mathbf{Q}} m(\mathbf{Q})\left(S^{exp}_{AE}(\mathbf{Q}) - S^{sim}_{AE}(\mathbf{Q})\right)^2. \quad (3)$$

This measure should be more robust to artifacts in both the experimental and simulation data. Indeed, as shown in Fig. 4b, it does slightly better in identifying an optimal $J_2$. The behavior of $\chi^2_{S_{AE}(\mathbf{Q})}$ as a function of the Hamiltonian parameters is similar to

the one obtained from the latent space representation of the $S(\mathbf{Q})$ data using a linear autoencoder, which is equivalent to the principal component analysis (PCA).

Here we propose an alternative error measure. The 30-dimensional latent space representation $S_L$ should, in some sense, capture the most relevant information in $S(\mathbf{Q})$. This suggests that to compare $S^{exp}(\mathbf{Q})$ to $S^{sim}(\mathbf{Q})$ we should actually look at the squared distance of their latent space vectors

$$\chi^2_{S_L} = \frac{1}{N_L}\sum_{L=1}^{D} \left(S^{exp}_L - S^{sim}_L\right)^2. \quad (4)$$

Figure 4c shows that this error measure produces the clearest minimum, and thus the most precise identification of $J_2$. We will use $\chi^2_{S_L}$ as our optimization cost function in what follows.

The inverse scattering problem for $Dy_2Ti_2O_7$ requires finding not just one, but three unknown Hamiltonian parameters: $J_2$, $J_3$, and $J_{3'}$. We employ a variant of the Efficient Global Optimization algorithm to find the Hamiltonian $H_{opt}$ that minimizes $\chi^2_{S_L}$ [24]. In this approach, one iteratively constructs a dataset of carefully sampled Hamiltonians $H$. For each Hamiltonian $H$, we calculate a simulated structure factor $S^{sim}(\mathbf{Q})$ and corresponding deviation $\chi^2_{S_L}$ from the experimental data. With all such data, one builds a Gaussian process regression model $\hat{\chi}^2_{S_L}(H)$ that predicts $\chi^2_{S_L}$ for Hamiltonians $H$ not yet sampled. The low-cost model $\hat{\chi}^2_{S_L}(H)$ can be rapidly scanned over the space of Hamiltonians. Also, $\hat{\chi}^2_{S_L}(H)$ acts as a denoiser, effectively "averaging out" uncorrelated stochastic errors in the $\chi^2_{S_L}$ data. As more data is collected, the improved models $\hat{\chi}^2_{S_L}(H)$ will progressively become more faithful

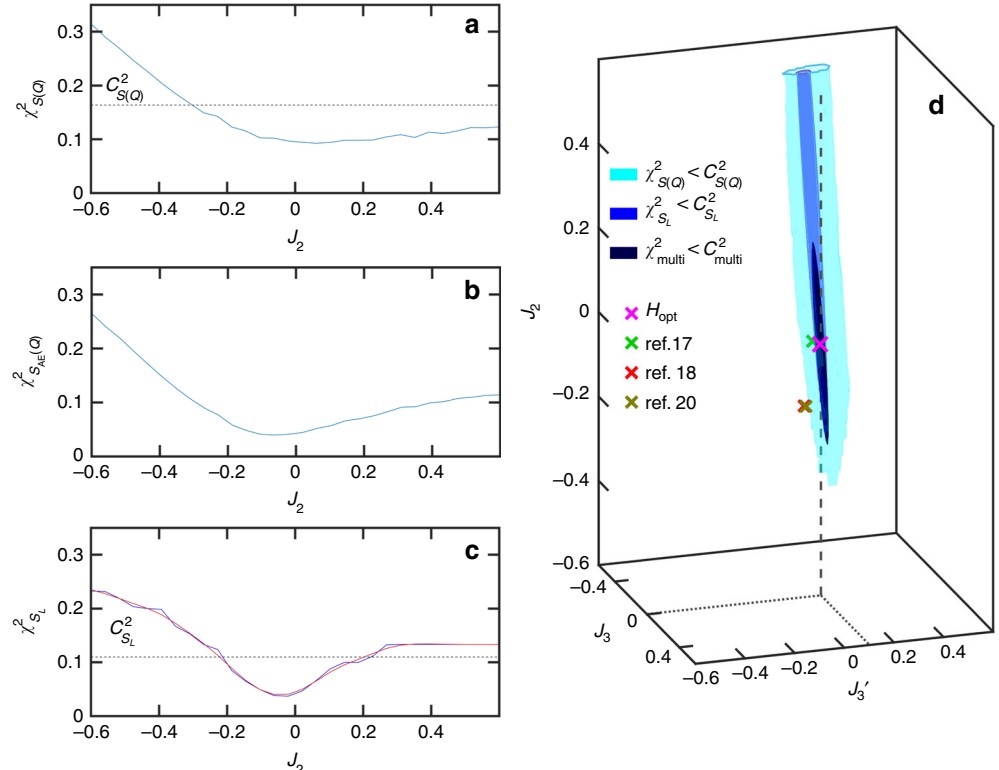

**Fig. 4 Cost functions and their effects.** Inferring the spin Hamiltonian for $Dy_2Ti_2O_7$. **a** $\chi^2_{S(\mathbf{Q})}$ directly measures the distance between experimental and simulated $S(\mathbf{Q})$ data. It is relatively flat and noisy around its minimum, thus yielding a large uncertainty in the $J_2$ coupling (any value below the dashed line, $C^2_{S(\mathbf{Q})}$, is a reasonable candidate). **b** $\chi^2_{S_{AE}(\mathbf{Q})}$ measures the distance between $S(\mathbf{Q})$ data after being filtered through the autoencoder. **c** $\chi^2_{S_L}$ measures the distance between the 30-dimensional latent space representations of the $S(\mathbf{Q})$ data. The Gaussian process model $\hat{\chi}^2_{S_L}(H)$(red curve) accurately approximates $\chi^2_{S_L}$, even in the full space of $J_2$, $J_3$, and $J_{3'}$. **d** Once a good model $\hat{\chi}^2_{S_L}(H)$ has been constructed, we can rapidly identify the optimal Hamiltonian model (magenta cross). The autoencoder-based error measure $\chi^2_{S_L}$ yields much smaller model uncertainty (blue region) than the naïve one $\chi^2_{S(\mathbf{Q})}$ (cyan region). Model uncertainty is further reduced using a multi-objective error measure $\chi^2_{multi}$ that incorporates heat capacity data (dark-blue region). Three most popular Hamiltonian sets currently used from refs. [17,18,20] have also been marked as green, red and gray crosses respectively.

to $\chi^2_{S_L}$. Optimization, as described in methods (Methods: Optimization) gives the optimal parameters as $J_2 = 0.34(6)$ K, $J_3 = -0.134(18)$ K and $J_{3'} = 0.102(32)$ K. The red curve of Fig. 4c shows a cross section of the final $\hat{\chi}^2_{S_L}(H)$ model. The minimum at $J_2 = 0$ is readily apparent. The dashed line in Fig. 4a indicates our empirically selected error tolerance threshold $C^2_{S(\mathbf{Q})}$. The dashed line in Fig. 4c shows $C^2_{S_L}$, the corresponding tolerance threshold for the latent space error. We calculated $C^2_{S_L}$ from $C^2_{S(\mathbf{Q})}$ under the assumption of a fixed amount of uncertainty in the scattering data [Methods: Uncertainty quantification]. Figure 4d shows the three-dimensional regions of uncertainty corresponding to $\chi^2_{S(\mathbf{Q})} < C^2_{S(\mathbf{Q})}$ (cyan) and $\chi^2_{S_L} < C^2_{S_L}$ (blue).

**Multi-modal optimization.** With more experimental constraints, we can further reduce uncertainty in $H_{opt}$. For this purpose, we define a new error measure $\chi^2_{multi} = \chi^2_{S_L} \times \chi^2_{c_v}$, where $\chi^2_{C_v} = \left(c_v^{exp} - c_v^{sim}\right)^2$ denotes the squared error between experimental[4] and simulated heat capacities, $c_v = \frac{1}{T^2}\left\langle U^2 - \langle U \rangle^2 \right\rangle$. Minimizing this multi-objective error function slightly modifies the model parameters: $J_2 = 0.00(6)$ K, $J_3 = -0.014(16)$ K and $J_{3'} = 0.102(16)$ K, pictured as a green cross in Fig. 4d. But perhaps more importantly, the uncertainties in these parameters have decreased significantly. This is illustrated by the very compact dark-blue region in Fig. 4d, for which $\chi^2_{multi} < C^2_{multi}$, where $C^2_{multi}$ is again calculated as a function of $C^2_{S(\mathbf{Q})}$.

The agreement between $S^{exp}(\mathbf{Q})$ and $S^{sim}_{opt}(\mathbf{Q})$ is quite good, as previously observed in Figs. 2 and 3. Further comparisons are shown in Supplementary Fig. 2. To truly validate the model, however, we should compare to experimental data that has not been used during the model optimization process. For this purpose, we use the magnetic field dependence of different physical properties shown in Fig. 5. The optimal spin model reproduces the measured field dependence of the magnetization[25], the zero-field cooled (ZFC) and field cooled (FC) magnetic susceptibility[12], and the diffuse scattering at multiple temperature and applied field conditions, confirming that we have indeed found a model Hamiltonian adequate to describe the magnetic properties of $Dy_2Ti_2O_7$ including the onset of irreversibility and glassiness.

## Discussion

Our present study has primarily focused on robust inference of the optimal model Hamiltonian. There are two important aspects of our methodology that we wish to emphasize. First, our use of an autoencoder, trained on large quantities of simulation data, provides a distance measure $\chi^2_{S_L}$ that allows robust comparisons to experimental scattering data. Second, our use of Gaussian process regression models $\hat{\chi}^2_{S_L}$ as a low-cost predictor for $\chi^2_{S_L}$ improves the quality of our optimized Hamiltonians. Gaussian process regression averages out uncorrelated stochastic error in $\chi^2_{S_L}$, and helps in making uncertainty estimates. The latter is crucial for guiding the design of future experiments or simulations. Whereas traditional analysis of diffraction and inelastic neutron scattering

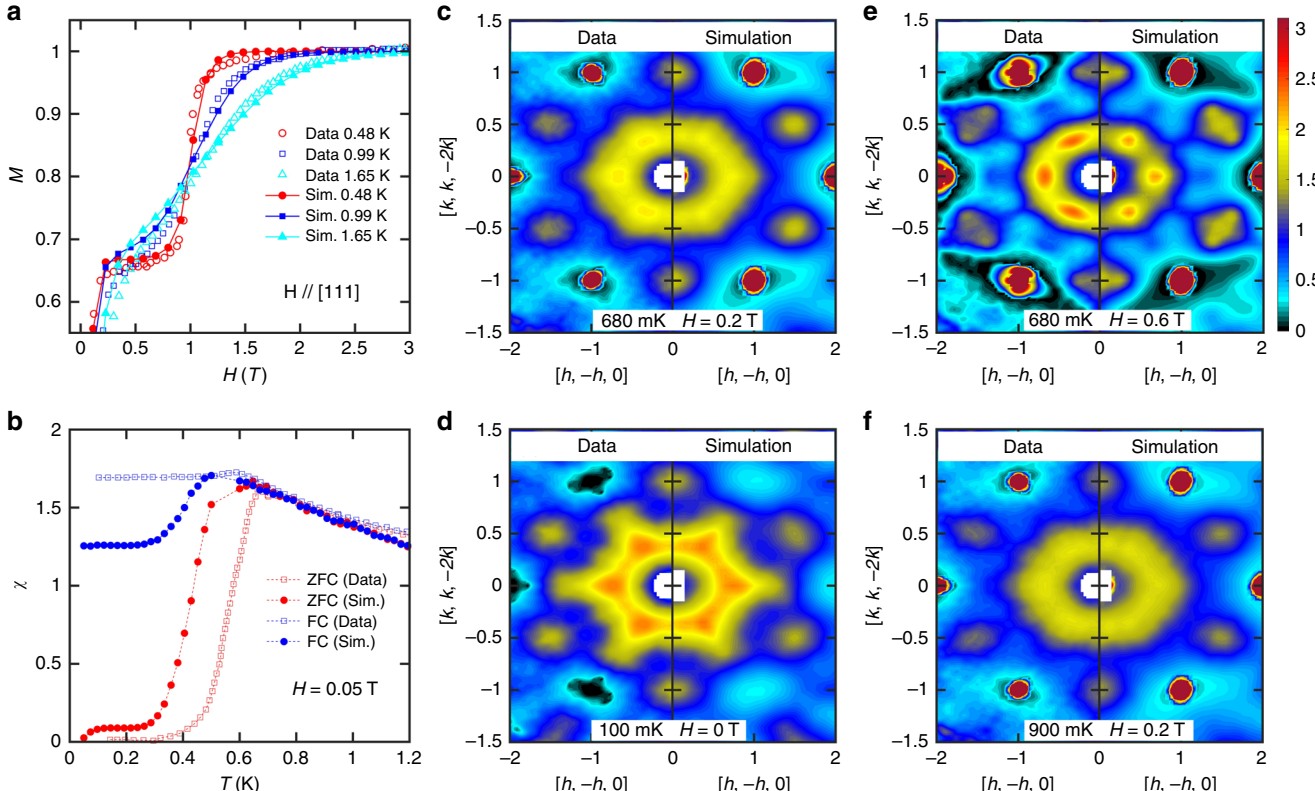

**Fig. 5 Validation of the optimal solution over multiple experiments. a** Magnetization as a function of magnetic field and temperature, **b** zero-field cooled (ZFC) – field cooled (FC) susceptibility, **c–f** magnetic diffuse scattering measured at different temperature and magnetic field combinations: [680 mK, 0.2 T], [100 mK, 0 T], [680 mK, 0.6 T] and [900 mK, 0.2 T] respectively. All the experiments and simulations shown here are done under magnetic field along the [1,1,1] direction. The magnetization and the ZFC-FC data are extracted from refs. [5,26], respectively.

is time consuming and error prone, our methodology is fully automated, and helps overcome difficulties of visualizing 3D or 4D data.

Finally, we remark that the autoencoder latent space provides an interesting characterization of structure factor data in its own right. Supplementary Fig. 5 in the supplement illustrates how the 30-dimensional latent space variables map to $S(\mathbf{Q})$. Supplementary Fig. 6 illustrates the activations of each latent space variable at varying points in the space of $J_3 − J_{3'}$ parameters.

Future studies might explore more direct application of autoencoders to the problems of background subtraction and of denoising experimental data. Here, we investigate another interesting application of the autoencoder: It can delineate different magnetic regimes. To demonstrate this, we will explore the space of $J_3$ and $J_{3'}$ parameters, while keeping $J_2 = 0$ K fixed. Our goal is to build a map of regimes with different dominant spatial magnetic correlations within this two-dimensional Hamiltonian space. We caution that the transitions between regimes will typically not be sharp phase transitions, so our modeling will not produce a phase diagram in the strict sense.

Figure 6a shows the result of our clustering analysis on the simulated data (Methods: Clustering). The optimal spin Hamiltonian for $Dy_2Ti_2O_7$ is marked as $H_{opt}$ near the center of this map. The corresponding $S^{sim}(\mathbf{Q})$ data, sliced in the high symmetry plane, had previously been shown in Fig. 3b. Figure 6b–i show the $S^{sim}(\mathbf{Q})$ data for alternative Hamiltonians, as marked on the map. It is clear from these results that the spin Hamiltonian of $Dy_2Ti_2O_7$ is close to the confluence of multiple regimes. This fact reveals an additional source of complexity that explains the difficulties that were encountered in previous characterizations of this material. This analysis suggests a roadmap for further

experimental studies. For example, the application of relatively small external fields and pressures or dopings should be enough to push $Dy_2Ti_2O_7$ into new magnetic regimes. For instance, the proximity to the ferromagnetic phase (blue regime in Fig. 6a) indicates that the saturation field is small, as confirmed by magnetization, Fig. 5a.

In summary, a fundamental bottleneck in experimental condensed matter physics is model optimization and assessment of confidence levels. We have developed a machine learning-based approach that addresses both challenges in an automated way. Applied to $Dy_2Ti_2O_7$ our method produces a model that accounts for the diffuse scattering data as well as the lack of magnetic ordering at low temperature. Our approach readily extends to the analysis of dynamical correlations, parametric data sets in e.g. field and temperature, and other scattering data.

## Methods

**Experimental details.** To measure the diffuse scattering of $Dy_2Ti_2O_7$ an isotopically enriched single crystal sample of $Dy_2Ti_2O_7$ was grown using an optical floating-zone method in a 5 atm oxygen atmosphere. Starting material $Dy_2O_3$ (94.4% Dy-162) and $TiO_2$ powder were first mixed in proper ratios and then annealed in air at 1400 °C for 40 h before growth in the image furnace as previously described[26]. Then the sample was further annealed in oxygen at 1400 °C for 20 h after the floating-zone growth. The lack of a nuclear spin moment in Dy-162 means that nuclear spin relaxation channels for the spins are cut off which is important in order to study the quench behavior in the material. In addition, the incoherent scattering from natural dysprosium is high (54.4 barns) whereas for Dy-162 it is zero and the absorption cross section is decreased from 994 barns (2200 ms$^{-1}$ neutrons) for natural dysprosium to 194 barns for isotope 162. A best growth was achieved with a pulling speed of 3 mm/hour. One piece of crystal with the mass ≈ 200 mg was aligned in the (111) plane for the neutron investigation at the single crystal diffuse scattering spectrometer CORELLI at the Spallation Neutron Source, Oak Ridge National Laboratory. The crystal was prepared as a sphere to minimize absorption corrections and demagnetization corrections.

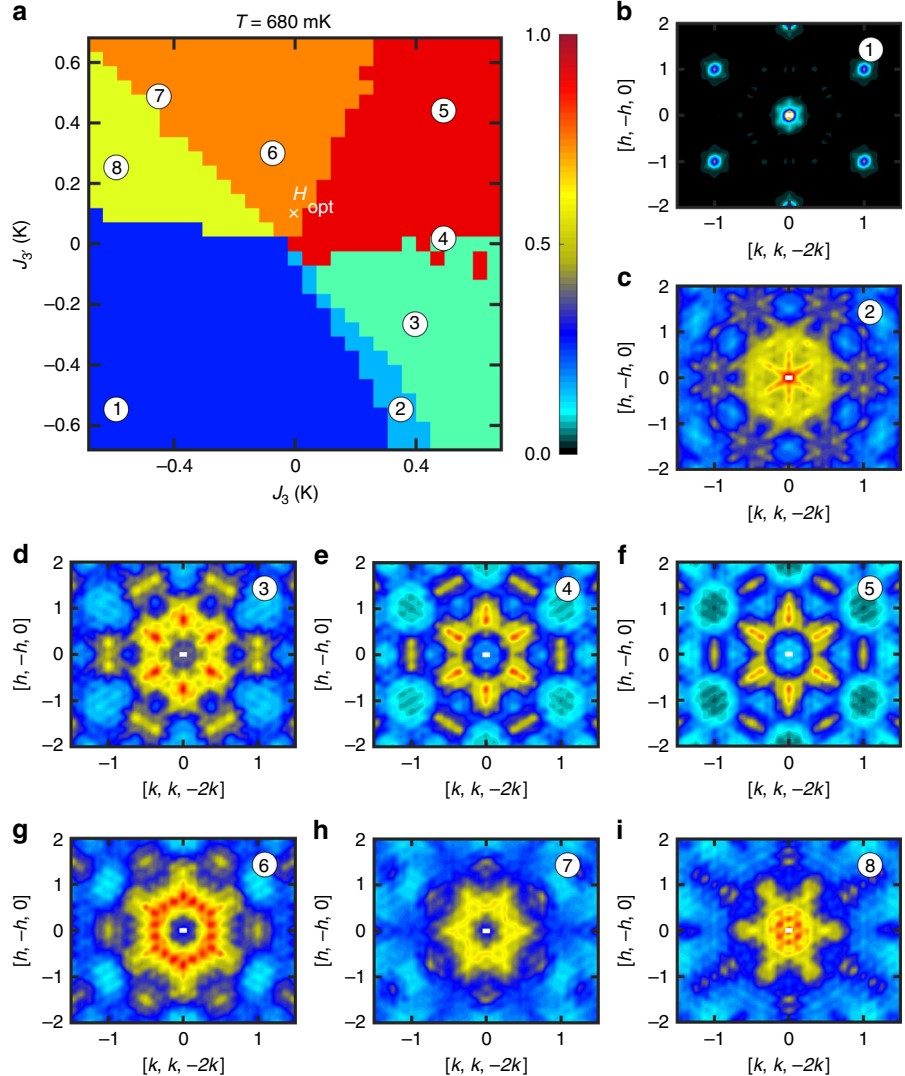

**Fig. 6 Map of regimes with different spatial magnetic correlations.** The map is generated by varying $J_3$ and $J_{3'}$, with the remaining Hamiltonian parameters $J_1$, $J_2$, and $D$ being fixed to their optimal values. The colors in **a** indicate different groups of $S(\mathbf{Q})$, which correspond to regimes with different patterns of magnetic correlations. **b–i** Simulated $S(\mathbf{Q})$ data sliced in the high symmetry plane at specific points indexed on panel (**a**). The large dark-blue regime (1) corresponds to long-range ferromagnetic order, as indicated by the Bragg peaks in panel (**b**). **c–i** Correspond to different patterns of short-range correlations arising from subsets of states that still obey ice rules. The pattern (**g**) falls within the same orange regime as $H_{opt}$, and qualitatively captures the Coulombic correlations expected for spin ice [cf. experimental scattering data in Fig. 3a].

CORELLI is a time-of-flight instrument where the elastic contribution is separated by a pseudo-statistical chopper[27]. The crystal was rotated through 180 degrees with the step of 5 degree horizontally with the vertical angular coverage of ±8 degree (limited by the magnet vertical opening) for survey on the elastic and diffuse peaks in reciprocal space. The dilution refrigerator insert and cryomagnet were used to enable the measurements down 100 mK and fields up to 1.4 T. The data were reduced using Mantid[28] and Python script available at Corelli. Background runs at 1.4 Tesla were made to remove all diffuse signal and the extra scattering at Bragg peak positions due to the polarized spin contribution was accounted for by using the zero field intensities. (see Supplementary Fig. 01) Figs. 2a and 3a shows a 3D plot and a slice of the high symmetry plane of the background-subtracted diffuse scattering measurement at 680 mK and 0 T respectively.

**Simulations details**. Given a model Hamiltonian $H$, we use Metropolis Monte Carlo to generate a simulated structure factor, $S^{sim}(\mathbf{Q})$, to be compared with the experimental data $S^{exp}(\mathbf{Q})$. We use simulated annealing to properly estimate $S^{sim}(\mathbf{Q})$[29]. Beginning at an initial temperature of 50 K, we iterate through 11 exponentially spaced intermediate temperatures, until finally reaching the target temperature of 680 mK. At each intermediate temperature, $5 \times 10^6$ Monte-Carlo sweeps were performed. At every sweep, each spin is updated once on average, according to the Metropolis acceptance criterion[30]. We perform our simulations

using $4 \times 4 \times 4$ cubic supercells, giving a total of 1024 spins in the pyrochlore lattice. The magnetic form factor of $Dy^{3+}$ and the neutron scattering polarization factor that enter in the calculation of the spin structure factor, $S(\mathbf{Q})$, are accounted before comparison to the background corrected experimental data. To correctly account for the long-range dipolar interactions, we used Ewald summation[31], implemented with the fast Fourier transform.

**Training details**. To train the autoencoder, we require a dataset sufficiently broad to cover all potentially important characteristic features of the $Dy_2Ti_2O_7$ scattering data. For this purpose, we employ 1000 model Hamiltonians of the form Eq. (1). Each model has individually randomized coupling strengths $J_2$, $J_3$, and $J_{3'}$, sampled uniformly from the range $-0.6$ K and $0.6$ K. For each model, we use simulated annealing to generate equilibrated three-dimensional $S^{sim}(\mathbf{Q})$ data at the target temperature of 680 mK. Our training data will thus consist of 1000 model Hamiltonians, each labeled by simulated data. The autoencoder tries to minimize the deviation between its input $S^{sim}(\mathbf{Q})$. and filtered output, summed over all random models in the dataset.

Training the autoencoder corresponds to determining the parameters (i.e., $W$, $b$, $W'$, and $b'$) that minimize a loss function $\mathcal{L}$. Primarily, we are interested in minimizing the squared error between the simulated data and autoencoder-filtered

output, summed over all models $H$ in the training dataset,

$$\mathcal{L} = \frac{1}{N} \sum_H \sum_\mathbf{Q} m(\mathbf{Q})(S(\mathbf{Q}) - S_{AE}(\mathbf{Q}))^2 + \frac{\lambda}{2} \sum_L \sum_\mathbf{Q} \left(w_\mathbf{Q}^L\right)^2 + \beta \sum_D KL(\rho||\hat{\rho}_D). \tag{5}$$

The second and third terms are relatively weak, and include two types of regularization: An $L_2$ regularization on the weight matrices $W$ and $W'$, and a sparsity regularization on the latent space activations $S_L$[32]. The sparsity regularization is a Kullback-Leibler divergence of average activation value, $\hat{\rho}_D$ of the hidden layer neurons and the desired average activation value, $\rho$ has been set to 0.05. The regularizer coefficients $\lambda$ and $\beta$ are set to 0.001 and 1, respectively. This regularization seems to improve the physical interpretability of the latent space representation. Despite having millions of trainable parameters in the neural network, the autoencoder does not seem prone to overfitting; the low-dimensionality of the latent space itself acts as a strong regularizer. To find the model parameters that minimize $\mathcal{L}$, we use the scaled conjugate gradient descent algorithm[33], as it is implemented in Matlab. We also experimented with a Keras autoencoder implementation, and found that it made little qualitative difference in our final results.

We found that a simple fully-connected autoencoder works well for experimental artifact removal from diffuse scattering data although other architectures can be explored in detail such as multilayer convolutional neural networks (CNN) or variational autoencoders. Note that artifact removal is inherent to our implementation of the AE due to the nature of the training data. Specifically, our dataset contained only simulated $S(\mathbf{Q})$ data, and the autoencoder is trained to reproduce that. Because experimental artifacts are not present in the simulated data, the autoencoder inherently filters them out.

**Optimization**. Optimization proceeds iteratively. We initially select 100 random Hamiltonians, where $J_2$, $J_3$, and $J_{3'}$ are each sampled uniformly from the range $-0.6$ K to 0.6 K. At each subsequent iteration, we use all available data to build $\hat{\chi}_{S_L}^2$, the low-cost approximator to $\chi_{S_L}^2$. Next, we randomly select 100 new Hamiltonians $H$ for inclusion in the dataset, each being sampled uniformly, subject to the constraint $\hat{\chi}^2(H) < c$. The cut-off parameter $c$ decreases exponentially, rescaling by a factor 0.9 at each iteration. Consequently, later iterations in the optimization procedure are focused on regions where $\chi_{S_L}^2$ is smallest. The optimization procedure terminates after about 40 iterations, at which point we take $H_{opt}$ to be the minimizer of $\hat{\chi}_{S_L}^2(H)$.

**Uncertainty quantification**. How can we compare uncertainties of $J_2$, as estimated from $\chi_{S(\mathbf{Q})}^2$ vs. $\chi_{S_L}^2$? From Fig. 4a alone, one might estimate that $J_2$ could lie anywhere between $-0.3$ K and 0.5 K. This is the region for which $\chi_{S(\mathbf{Q})}^2 < C_{S(\mathbf{Q})}^2$, where $C_{S(\mathbf{Q})}^2$ is an empirically selected tolerance denoted by the dashed horizontal line. Working backwards, we can then ask: How much noise in the simulated $S^{sim}(\mathbf{Q})$ data would it take for $C_{S(\mathbf{Q})}^2$ to be the actual stochastic uncertainty in $\chi_{S(\mathbf{Q})}^2$? Assuming that $S^{sim}(\mathbf{Q})$ contains this level of noise magnitude, we can measure the corresponding stochastic uncertainty $C_{S_L}^2$ of $\chi_{S_L}^2$, which we plot as the dashed line in Fig. 4c. Comparing with Fig. 4a, we conclude that the autoencoder-based error measure $\chi_{S_L}^2$ is more robust to stochastic noise, i.e., allows more precise estimation of $J_2$. Thus, we have selected $\chi_{S_L}^2$ as the best cost function for inferring the model Hamiltonian from the structure factor data.

Given experimental heat capacity data $c_v$, we introduced a multi-objective cost function $\chi_{multi}^2$. Repeating the same procedure as above, we can define the multi-objective tolerance threshold $C_{multi}^2$ in terms of the raw tolerance $C_{S(\mathbf{Q})}^2$.

**Clustering**. To determine magnetic regimes, we employ the agglomerative hierarchical clustering algorithm[34]. For this, we use the same dataset as was used to train the autoencoder, i.e., a random selection of 1000 model Hamiltonians, and their corresponding $S^{sim}(\mathbf{Q})$ data. The clustering algorithm requires as input the pairwise distances between all points in the dataset. We again employ the squared distance in the autoencoder latent space, i.e., as it appeared in $\chi_{S_L}^2$.

## Data availability
The data that support the findings of this study are available from the corresponding author upon reasonable request.

## Code availability
The computer codes that support the finding of this study are available from the corresponding author upon reasonable request.

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

## Acknowledgements

A.M.S., Q.Z., and F.Y. acknowledge the support from the US DOE office of scientific user facilities. Z.L.D and H.D.Z thank the NSF for support with grant number DMR-1350002. A portion of this research used resources at Spallation Neutron Source, a DOE Office of Science User Facility operated by the Oak Ridge National Laboratory. The research by D.A.T. was sponsored by the DOE Office of Science, Laboratory Directed Research and Development program (LDRD) of Oak Ridge National Laboratory, managed by UT-Battelle, LLC for the U.S. Department of Energy. (Project ID 9566). Support for Q.Z. was provided by US DOE under EPSCoR Grant No. DESC0012432 with additional support from the Louisiana board of regent. K.B. acknowledges support from the LDRD program at Los Alamos National Laboratory. Y.W.L., M.E., and resources for computer modeling are sponsored by the Oak Ridge Leadership Computing Facility, which is supported by the Office of Science of the U.S. Department of Energy under contract no. DE-AC05-00OR22725. D.A.T. and A.M.S. would like to thank Guannan Zhang for useful discussions.

## Author contributions

D.A.T. conceived and coordinated the project. H.Z. and Z.L.D. prepared and characterized the high-quality single crystal. A.M.S., D.A.T, F.Y., and Q.Z. performed the neutron experiments. A.M.S., D.A.T., and Q.Z. undertook the initial data analysis. A.M.S. performed the numerical simulations and machine learning analysis with input and guidance from D.A.T., C.D.B., K.B., Y.W.L, S.A.G., and M.E. D.A.T., A.M.S., C.D.B., K.B., and S.A.G. wrote the paper. V.S. contributed in double checking and verifying ML results. All the authors discussed the data and its interpretation.

## Competing interests

The authors declare no competing interests.
