## [Peer Review File · Nature Communications]

Reviewers' comments:

Reviewer #1 (Remarks to the Author):

In this paper, the authors proposed a method using a machine learning technique to extract the parameters in the microscopic Hamiltonian for the spin ice material Dy₂Ti₂O₇ from diffused neutron scattering data. This is an interesting proposal that takes advantage of recent progress of machine learning in order to obtain crucial information about a specific material from experimental data.

Reverse engineering from experiments to find the optimal Hamiltonian to describe Dy₂Ti₂O₇ has been attempted before using experimental data from scattering and thermodynamic measurements (Ref. [18]); however, it is not easy to judge the quality of estimated interactions. The major difficulty in this reverse problem is the uncertainties due to disorder in the material and noises in the data allow a big range of parameter sets to generate very similar patterns. Here, the authors trained an autoencoder neural network to extract the key features in the data, and use the features in the latent space as a guide to obtain the optimal Hamiltonian. This technique has been used in image processing for feature extraction and noise reduction, and its application to processing scattering data seems a natural extension.

There are a couple things that I hope the authors can clarify:

1. One important thing about the autoencoder is to understand the information compressed in the latent variables. The authors argue that χ_{S_L} is the best cost function for optimization because of the appearance of a clear minimum; however, I would also emphasize that the fact χ_{S_L} can be accurately approximated by the Gaussian process model indicating the latent variables are independent, and the essential information is compressed. This behavior is very similar to what the variational autoencoder aims to achieve. Although the authors in the last section of the main text discuss the usage of latent variables as a means to define the distance between different S(Q)'s, it would be interesting to make connection to the activation function output in Fig. S6.
2. The last sentence in the caption of Fig. 3 can be misleading: "(d)-(f) With just 1, 6, and 12 latent space components, the autoencoder can reasonably capture important characteristics ...". However in the main text, it is stated, "we replace 1, 6, and 12 latent space components of S_L with the corresponding ones in S_L'." These two sentences don't convey exactly the same meaning.
3. What criterion is used to determine that 30 latent variables are enough to compress the scattering data? It is not clear from either the main text or the method section.
4. The authors should provide more details on the specifications of the network architecture, hyper-parameters for training, and any data pre-processing methods for future reproduction of their results.
5. The authors used C_v, magnetization, and susceptibility to validate the model Hamiltonian. I wonder if the model Hamiltonian can also give accurately the liquid-gas transition line for Dy₂Ti₂O₇ in [111] field, as in Fig. 4 in Ref. [3]?

In conclusion, I believe this paper provides a new direction using ML methods to extract the interaction strength in the microscopic Hamiltonian for Dy₂Ti₂O₇ directly from the scattering data. I recommend the publication of the paper after the above points are properly addressed.

Typo:

1. The last sentence in the caption of Fig. 5 appeared twice.

Reviewer #2 (Remarks to the Author):

This article details a new and novel method to calculate/simulate a magnetic exchange Hamiltonian from neutron scattering data using a method of training an autoencoder to learn to represent different signals. Much of today's magnetic modelling involves approximated systems constrained to a particular Hamiltonian model with defined and limited parameters. Exploring all of this parameter space is infeasible with the current methods. However this group introduces a new method which remains robust even when faced with a complicated set of interactions and inputs. The authors claim that their method can address such issues as background subtractions and noise giving more reliable simulations.

I have some questions/comments that I would like to see answered:

- (1) Measurements on DTO were taken in zero field and 1.4T when the system enters a field-induced long range ordered state. The data were manipulated to include features from both data sets to clarify the diffuse magnetic scattering. Does the autoencoder method allow for the modelling of different applied field states in a system – for instance the plateau states in $\text{SrCu}_2(\text{BO}_3)_2$?
- (2) What is the effect of the magnetic structure on the method of autoencoder? In some methods such as linear spin wave theory, an assumption is made that the system is magnetically ordered. Is this considered at all? What would the effect of this have on the learning process?
- (3) What was the computing time required to train the autoencoder? Is this something that requires specialised computing clusters or something that can eventually be run on a PC or laptop?
- (4) How does this program distinguish between relevant and irrelevant noise in the weak diffuse scattering?
- (5) Is this method transferable to $S=1/2$ quantum magnets where zero point spin fluctuations play a significant role in the competing interactions?

Additional points to address:

In Fig, 5: Validation of the optimal solution – please check the lettering in the figure caption. (b) looks to be (d) and b,c and d all need to be mentioned in the caption

Supplemental section:

Fig S1: The caption here is unclear – when it says “The overall non-magnetic background (d) has been determined by replacing the Bragg peak intensity measured at 1.4T by the zero field values.” Was it only the peak shape that was swapped? How far away from the Bragg peaks did the replacement go as Fig S1b shows some over subtraction? It also appears that the sample-mount powder lines were also subtracted to help reveal more details. Perhaps some circles as to the feature that were actually exchanged in this process could be overlaid on the figures? Was it (a) – (c) = (d) – and then (d) was subtracted from (a) to give (b)?

Fig S6: What is the colour scale for these plots? Is it the same as S5?

This is a very well written article and with a novel approach to modelling that I believe will benefit many in the magnetism community. Therefore, after the authors consider and respond to these comments I would like to recommend publication.

Reviewer #3 (Remarks to the Author):

The manuscript by Samarakoon et al. describes how the use of a machine-learning autoencoder as a data filtering/processing tool can enhance the analysis of neutron scattering data. In particular, the encoder acts as a data filter, while the bottleneck layer in the network acts as a convenient space for performing similarity analysis and error calculation in a way that has abstracted away spurious aspects of the data and simulation (such as noise). This allows them to more easily identify the correct Hamiltonian, and explore variations thereof.

This work appears to be technically sound. This is a timely contribution given the rising popularity of machine-learning methods in the analysis of scientific data. While many groups seem to be deploying ML for the sake of it, the present contribution instead is a meaningful and thoughtful

application of ML methods, where the advantages of ML are brought to bear to improve a particular data analysis. I commend the authors on this quality contribution.

Overall, I find this to be a strong contribution and certainly worthy of publication.

I have a number of minor comments which the authors may wish to consider:

1. The authors implement a very simple auto-encoder (fully connected, single hidden layer). Can the authors explain their rationale for doing so? A multi-layered convolutional neural network (CNN) is another obvious candidate, and has been applied with enormous success for AE and generating latent spaces for (e.g.) image data. While a CNN adds some complexity, the locality of a CNN is a natural fit to things like images and scattering data that have considerable locality in the distribution of the data. Did the authors try using a CNN? Was the choice of a simple method for reasons of computational cost? Implementation simplicity? Something else? A simple sentence explaining the rationale would suffice.

2. The authors note (e.g. bottom of page 6) that the AE removes artifacts. This of course makes sense because the bottleneck aspect of the AE will throw away any non-pertinent aspects of the data. However the authors also note using a mask when preprocessing their data. Thus I am left wondering whether the artifact removal is simply a matter of these artifacts having been masked (and thus unavailable to the AE and instead filled in by the AE based on training data), or whether the artifact removal is 'inherent' to the AE (because of the natural coarse-graining and heuristic nature of the AE). On a related note, I wonder what the effect of the AE on high-resolution features in the input data? If there are "real" (physically meaningful) high-resolution features in the input data, will the AE preserve these? Or will they become washed out? (As the authors are probably aware, AE very frequently lead to 'fuzzy' or 'blurry' versions of the input data. This is very noticeable in images but may be harder to see in the provided scattering images.) Can the authors quantify the q-resolution that is preserved after AE filtering?

3. The authors note (page 7, and Figure 3) that there is a small change for a "relatively significant change to J_2 ". However it is hard to judge how big of a J_2 change this is without being told the absolute J_2 value. The authors should note the starting value of J_2 (or explain the % change that the applied modification represents) so that readers can better judge for themselves whether this is a "significant" change.

4. On page 8 (and Equation 3): The authors compare the error (chi-squared) computed from data/sim to the autoencoder-filtered versions, and to the distance in the latent space. They demonstrate that the distance in the latent space is the cleanest metric, since this space abstracts away irrelevant details and encodes only the most useful information. It thus makes perfect sense that this space would be robust and noise-free. However, one can also ask the question as to why there is actually a difference between the latent space method and the filtered $S(Q)$ method. There should be a one-to-one correspondence from a particular latent space point to its unique filtered $S(Q)$. So why are the two any different? I can of course think of reasons. For instance, the decoder part of the AE has network weights selected to reproduce training data, which includes noise. Thus the decoder network weights have some spurious component (e.g. in the least significant bits of any given weight) that generate some random variability in the outputs. This re-introduces noise and thus contaminates the reconstructed (filtered) $S(Q)$, and makes the error measure less robust. Thus the bottleneck layer is the ideal representation for computing distances/errors. However perhaps there is some other reason why there is a difference? Did the authors do any tests to determine the origin of this difference in the spaces? The authors may want to have a sentence

that explains why they see this difference between the spaces.

5. A very small point is where the authors write (page 9) "Here we propose an alternative, and perhaps less obvious, error measure."; it is subjective what is more or less obvious. In reading the paper my immediate thought was to use the latent space as the distance-measurement space. I believe most AI researchers would similarly view it most natural to measure similarity (and thus error) in the latent space, since this space is the "minimal conceptual representation of the problem" learned by the system. However this is likely not obvious to scientists new to machine learning methods, and thus would not be obvious to them. This is just a comment; I don't think the authors need to modify this text.

6. Page 3, "increases drastically" is ambiguous. Increases with respect to what? (Temperature?)

7. The caption for Figure 3 says "(d)-(f) With just 1, 6, and 12 latent space components, the autoencoder can reasonably capture important characteristics of ΔS^{sim} .", which sounds like the autoencoder was retrained with a latent space (bottleneck vector) of only length 1, 6, or 12. However from the main text it is clarified that what was actually done was to modify 1, 6, or 12 of the 30 features of the latent vector. The caption should be clarified.

Response to referees, NCOMMS-19-25545-T

We thank the referees for the encouraging and thoughtful remarks. After careful consideration, we have made a few revisions to the manuscript. Below, we respond to the referees' comments individually. Note that new text is in red color in the revised manuscript.

Reviewer #1

There are a couple things that I hope the authors can clarify:

1. One important thing about the autoencoder is to understand the information compressed in the latent variables. The authors argue that χ_{SL} is the best cost function for optimization because of the appearance of a clear minimum; however, I would also emphasize that the fact χ_{SL} can be accurately approximated by the Gaussian process model indicating the latent variables are independent, and the essential information is compressed. This behavior is very similar to what the variational autoencoder aims to achieve. Although the authors in the last section of the main text discuss the usage of latent variables as a means to define the distance between different $S(Q)$'s, it would be interesting to make connection to the activation function output in Fig. S6.

Indeed, our central point is that the latent space is, in effect, compressing the essential information of the $S(Q)$ data. Figures S5 and S6 help to illustrate the latent space variables. Although it is difficult for us to make a conclusive statement regarding their interpretation, we agree that it is important to remark on them in the context χ^2_{SL} . We have added a reference to these supplemental figures near the end of the manuscript (paragraph starting with "Finally, we remark that the autoencoder latent space provides an interesting characterization of structure factor data in its own right.")

2. The last sentence in the caption of Fig. 3 can be misleading: "(d)-(f) With just 1, 6, and 12 latent space components, the autoencoder can reasonably capture important characteristics ...". However, in the main text, it is stated, "we replace 1, 6, and 12 latent space components of S_L with the corresponding ones in S'_L ." These two sentences don't convey exactly the same meaning.

We thank the reviewer for pointing out this confusing text. The revised caption in Fig. 3 now states: "(d)-(f) The change in the autoencoder output when 1, 6, and 12 latent space components, respectively, are updated to account for the perturbation on J_2 (see main text for details)."

3. What criterion is used to determine that 30 latent variables are enough to compress the scattering data? It is not clear from either the main text or the method section.

We added three sentences of discussion following the point where the 30-dimensional latent space is introduced. In particular, we explain how $D=30$ is selected by finding the right balance between overfitting and underfitting.

Sentence: “The dimension D of the latent space should strike a balance between overfitting and underfitting. Keeping D relatively small limits the autoencoder’s ability to fit irrelevant noise in the training data. On the other hand, D should be large enough to allow the autoencoder flexibility to capture physically relevant characteristics in $S(\mathbf{Q})$. After some trial and error, we selected $D = 30$. (see Fig. S7)”

4. The authors should provide more details on the specifications of the network architecture, hyper-parameters for training, and any data pre-processing methods for future reproduction of their results.

We fully agree with the sentiment. As mentioned above, we have added a discussion to the manuscript about our choice to use a $D=30$ -dimensional latent space. We have also added details about the L_2 weight regularization and the sparsity regularization.

Sentence:

$$\mathcal{L} = \frac{1}{N} \sum_H \sum_Q m(\mathbf{Q}) (S(\mathbf{Q}) - S_{\text{AE}}(\mathbf{Q}))^2 + \frac{\lambda}{2} \sum_L \sum_Q (w_Q^L)^2 + \beta \sum_D KL(\rho || \hat{\rho}_D) .$$

The second and third terms are relatively weak, and include two types of regularization: An L_2 regularization on the weight matrices W and W' , and a sparsity regularization on the latent space activations S_L [32]. The sparsity regularization is a Kullback-Leibler divergence of average activation value, $\hat{\rho}_D$ of the hidden layer neurons and the desired average activation value, ρ has been set to 0.05. The regularizer coefficients λ and β are set to 0.001 and 1 respectively.”

Note that we used scaled conjugate gradient descent algorithm to globally optimize the model parameters. Thus, there are essentially no hyperparameters to select for the training procedure. This is unlike stochastic gradient descent training, which typically involves hyperparameters for the timestep, the momentum, the minibatch size, the early stopping criteria, etc. None of these are relevant to us.

We used only data pre-processing methods that are standard in neutron scattering. For example, Fig. S1 illustrates our application of background subtraction to the raw scattering data. Explaining this methodology in detail would be out of scope for this work, but hopefully its absence does not produce a barrier to reproducibility.

5. The authors used C_v , magnetization, and susceptibility to validate the model Hamiltonian. I wonder if the model Hamiltonian can also give accurately the liquid-gas transition line for $\text{Dy}_2\text{Ti}_2\text{O}_7$ in $[111]$ field, as in Fig. 4 in Ref. [3]?

The principal features of the liquid-gas transition with field along $[111]$ are dominated by the nearest neighbor and dipolar terms, which are unchanged in our model compared with previous models such as references [3] or [18]. An indication of this agreement is seen in the magnetization traces shown in Fig. 5a for three different temperatures as a function of H along $[111]$ traversing the transition line. The addition of further neighbor interactions has an effect on the angular dependence of this transition as the field is tilted away from $[111]$. A detailed study of this angular dependence has been presented in Ref. [17]. The parameters of our model Hamiltonian fall within the range determined on Ref [17] to reproduce the experimental results. We believed that this discussion is a bit specialized and would deviate too far from the point of this work and have thus opted not to include it.

1. The last sentence in the caption of Fig. 5 appeared twice.

This is now corrected, thank you and now it reads

“Validation of the optimal solution over multiple experiments: (a) Magnetization as a function of magnetic field and temperature, (b) zero-field cooled (ZFC) – field cooled (FC) susceptibility, (c)-(f) magnetic diffuse scattering measured at different temperature and magnetic field combinations: [680 mK, 0.2 T], [100 mK, 0 T], [680 mK, 0.6 T] and [900 mK, 0.2 T] respectively. All the experiments and simulations shown here are done under magnetic field along the $[1,1,1]$ direction. The magnetization and the ZFC-FC data are extracted from Refs. [26] and [5] respectively.”

Reviewer #2

(1) Measurements on DTO were taken in zero field and 1.4T when the system enters a field-induced long range ordered state. The data were manipulated to include features from both data sets to clarify the diffuse magnetic scattering. Does the autoencoder method allow for the modelling of different applied field states in a system – for instance the plateau states in $\text{SrCu}_2(\text{BO}_3)_2$?

In principle the autoencoder can be trained for applied fields and temperatures and so used for optimizing parameters from such data. It can also be trained for inelastic data too which has a richer information content than Bragg peaks from order. We have successfully used our methods on dynamics now.

We have modified the final sentence of the manuscript to “Our approach readily extends to the analysis of dynamical correlations, parametric data sets in e.g. field and temperature, and other scattering data.”

(2) What is the effect of the magnetic structure on the method of autoencoder? In some methods such as linear spin wave theory, an assumption is made that the system is magnetically ordered. Is this considered at all? What would the effect of this have on the learning process?

We do not need to assume the presence or absence of magnetic ordering, since this a possible outcome of the simulation. As it is shown for the case of dysprosium titanate, the MC simulations can output magnetic ordering in some region of the Hamiltonian space. This magnetic ordering produces sharp features in the diffuse scattering data that the autoencoder will automatically include among the minimum set of features required to better discriminate between different Hamiltonians.

(3) What was the computing time required to train the autoencoder? Is this something that requires specialized computing clusters or something that can eventually be run on a PC or laptop?

Once the data is available, the autoencoder can be trained in minutes. However, collecting the simulated $S(\mathbf{Q})$ data for thousands of model Hamiltonians took hundreds of CPU core-hours.

(4) How does this program distinguish between relevant and irrelevant noise in the weak diffuse scattering?

The autoencoder considers information “relevant” if it can find a statistically significant pattern from the large collection of simulated data. Everything else gets filtered out.

(5) Is this method transferable to $S=1/2$ quantum magnets where zero-point spin fluctuations play a significant role in the competing interactions?

If we restrict to the classical MC method that we are using for solving the direct problem, the answer is *yes*, as long as the temperature of the experiment is higher than the quantum to classical crossover temperature, so the diffuse scattering data can be simulated with classical methods (see <https://journals.aps.org/prb/abstract/10.1103/PhysRevB.96.134408>, <https://iopscience.iop.org/article/10.1088/1742-5468/2008/05/P05017>). If this is not the case, the direct problem must be solved by using methods, such as quantum MC or DMRG, that can account for the role of quantum fluctuations. One can still use the same protocol (including the

autoencoder) that is described in the manuscript. The only potential limitation is the *efficiency* of the method that is used for solving the direct problem (simulation of the quantum mechanical model).

In Fig. 5: Validation of the optimal solution – please check the lettering in the figure caption. (b) looks to be (d) and b,c and d all need to be mentioned in the caption

Figure 05 is corrected for lettering.

Fig S1: The caption here is unclear – when it says “The overall non-magnetic background (d) has been determined by replacing the Bragg peak intensity measured at 1.4T by the zero field values.” Was it only the peak shape that was swapped? How far away from the Bragg peaks did the replacement go as Fig S1b shows some over subtraction? It also appears that the sample-mount powder lines were also subtracted to help reveal more details. Perhaps some circles as to the feature that were actually exchanged in this process could be overlaid on the figures? Was it $(a) - (c) = (d)$ – and then (d) was subtracted from (a) to give (b) ?

It is not $(a) - (c) = (d)$ and $(b) = (a) - (d)$. We have added more explanation of manual data pre-processing starting from the sentence “Since the magnetic system...” on the page 2 of supplementary information for clarity.

Fig S6: What is the colour scale for these plots? Is it the same as S5?

A color scale for Fig. S5 is added now for clarity.

Reviewer #3 (Remarks to the Author):

1. The authors implement a very simple auto-encoder (fully connected, single hidden layer). Can the authors explain their rationale for doing so? A multi-layered convolutional neural network (CNN) is another obvious candidate, and has been applied with enormous success for AE and generating latent spaces for (e.g.) image data. While a CNN adds some complexity, the locality of a CNN is a natural fit to things like images and scattering data that have considerable locality in the distribution of the data. Did the authors try using a CNN? Was the choice of a simple method for reasons of computational cost? Implementation simplicity? Something else? A simple sentence explaining the rationale would suffice.

We selected a very simple auto-encoder for simplicity. CNNs would be another good choice, and are especially powerful for computer vision tasks. The great advantage of CNNs is that the convolution operator processes the image plane in a translation-invariant way. While this may be the right thing for computer vision tasks (e.g., “recognize a cat no matter where it appears in the image), it may not be optimal for processing $S(\mathbf{Q})$ data which lacks translation invariance in \mathbf{Q} -space. For example, for the class of Hamiltonian models we’re studying, the structure factor peaks always appear in specific locations. Nonetheless, we agree that CNNs are an interesting idea for future research.

2. The authors note (e.g. bottom of page 6) that the AE removes artifacts. This of course makes sense because the bottleneck aspect of the AE will throw away any non-pertinent aspects of the data. However, the authors also note using a mask when preprocessing their data. Thus, I am left wondering whether the artifact removal is simply a matter of these artifacts having been masked (and thus unavailable to the AE and instead filled in by the AE based on training data), or whether the artifact removal is 'inherent' to the AE (because of the natural coarse-graining and heuristic nature of the AE).

Artifact removal is ‘inherent’ to our implementation of the AE *due to the nature of the training data*. Specifically, our dataset contained only simulated $S(\mathbf{Q})$ data, and the autoencoder is trained to reproduce that. Because experimental artifacts will not be present in the simulated data, the autoencoder inherently filters them out.

On a related note, I wonder what the effect of the AE on high-resolution features in the input data? If there are "real" (physically meaningful) high-resolution features in the input data, will the AE preserve these? Or will they become washed out? (As the authors are probably aware, AE very frequently lead to 'fuzzy' or 'blurry' versions of the input data. This is very noticeable in images but may be harder to see in the provided scattering images.) Can the authors quantify the q-resolution that is preserved after AE filtering?

This is an interesting question. Indeed, CNNs tends to wash out high-resolution features, such as structure factor peaks. Our model is not a CNN, however, and that turns out to be an advantage in this case. Whereas CNNs employ “local” convolutional filters, our model makes no assumptions about locality in \mathbf{Q} -space. Therefore, if the $S(\mathbf{Q})$ data in the training set consistently exhibits sharp peaks, at precise \mathbf{Q} -vectors, then our model should be able to reproduce these peaks without any blurring.

3. The authors note (page 7, and Figure 3) that there is a small change for a "relatively significant change to J_2 ". However, it is hard to judge how big of a J_2 change this is without being told the absolute J_2 value. The authors should note the starting value of

J_2 (or explain the % change that the applied modification represents) so that readers can better judge for themselves whether this is a "significant" change.

We agree. Given that we are referring to the effective error bar of one of the exchange parameters (J_2), it is most clear to refer to the change in this variable as a percentage of the dominant one ($J_1=1.7K$). We are varying J_2 between $-0.08K$ and $0.09K$, which corresponds to $\pm 5\% J_1$. The revised figure caption now states this clearly.

Revised figure caption: “(c) The change in the simulated structure factor ΔS^{sim} when the J_2 coupling is perturbed from $0.16 K$ to $-0.18 K$. The relatively weak response ΔS^{sim} to a perturbation in J_2 of $\pm 0.05 J_1$ illustrates the challenge in inferring the correct spin Hamiltonian. (d)-(f) The change in the autoencoder output when 1, 6, and 12 latent space components, respectively, are updated to account for the perturbation on J_2 “

4. On page 8 (and Equation 3): The authors compare the error (chi-squared) computed from data/sim to the autoencoder-filtered versions, and to the distance in the latent space. They demonstrate that the distance in the latent space is the cleanest metric, since this space abstracts away irrelevant details and encodes only the most useful information. It thus makes perfect sense that this space would be robust and noise-free. However, one can also ask the question as to why there is actually a difference between the latent space method and the filtered $S(Q)$ method. There should be a one-to-one correspondence from a particular latent space point to its unique filtered $S(Q)$. So why are the two any different? I can of course think of reasons. For instance, the decoder part of the AE has network weights selected to reproduce training data, which includes noise. Thus, the decoder network weights have some spurious component (e.g. in the least significant bits of any given weight) that generate some random variability in the outputs. This re-introduces noise and thus contaminates the reconstructed (filtered) $S(Q)$, and makes the error measure less robust. Thus, the bottleneck layer is the ideal representation for computing distances/errors. However perhaps there is some other reason why there is a difference? Did the authors do any tests to determine the origin of this difference in the spaces? The authors may want to have a sentence that explains why they see this difference between the spaces.

This is an excellent question. To be honest, we're not really sure why there should be a big difference between the "latent space method" and the "filtered $S(Q)$ method". We tried both and empirically we found the latter to perform moderately better. Compare, e.g., panels (b) and (c) in Fig. 4. We find the Reviewer's thoughts about the bottleneck idea intriguing, and worthy of study in future work.

Incidentally, after reading the referee's insightful comment, we realized that the distance in latent space is very sensitive to the orthogonality (or lack thereof) of the AE weight matrix \mathbf{W} . For future work, it would be very interesting to apply an explicit orthogonality regularization along the lines of this blog post:

<https://towardsdatascience.com/build-the-right-autoencoder-tune-and-optimize-using-pca-principles-part-i-1f01f821999b>

5. A very small point is where the authors write (page 9) "Here we propose an alternative, and perhaps less obvious, error measure."; it is subjective what is more or less obvious.

We agree and have removed "and perhaps less obvious".

6. Page 3, "increases drastically" is ambiguous. Increases with respect to what? (Temperature?)

This is correct. The revised text now says: "increases drastically upon lowering the temperature leading to irreversible..."

7. The caption for Figure 3 says "(d)-(f) With just 1, 6, and 12 latent space components, the autoencoder can reasonably capture important characteristics of ΔS^{sim} .", which sounds like the autoencoder was retrained with a latent space (bottleneck vector) of only length 1, 6, or 12. However from the main text it is clarified that what was actually done was to modify 1, 6, or 12 of the 30 features of the latent vector. The caption should be clarified.

We agree. The revised caption now says "The change in the autoencoder output when 1, 6, and 12 latent space components, respectively, are updated to account for the perturbation on J_2 (see main text for details)."

REVIEWERS' COMMENTS:

Reviewer #1 (Remarks to the Author):

In this revised version of the manuscript and the reply to the referees, the authors have properly addressed all the referees' comments regarding the previous version. Especially, clarifications on the details of the autoencoder and data processing make it possible for future applications of the method. Therefore, I recommend this manuscript for publication.

Reviewer #2 (Remarks to the Author):

I am satisfied that the comments of all 3 reviewers have been answered satisfactorily. I advise publishing in its current form.

Reviewer #3 (Remarks to the Author):

The authors have adequately responded to the issues raised by all the reviewers. I believe the manuscript is suitable for publication in its present form.